

# Comprehensive analysis of long non-coding RNAs and mRNAs in skeletal muscle of diabetic Goto-Kakizaki rats during the early stage of type 2 diabetes

Wenlu Zhang,  Yunmeng Bai,  Zixi Chen,  Xingsong Li,  Shuying Fu, Lizhen Huang,  Shudai Lin and  Hongli Du

School of Biology and Biological Engineering, South China University of Technology, Guangzhou, China

## ABSTRACT

Skeletal muscle long non-coding RNAs (lncRNAs) were reported to be involved in the development of type 2 diabetes (T2D). However, little is known about the mechanism of skeletal muscle lncRNAs on hyperglycemia of diabetic Goto-Kakizaki (GK) rats at the age of 3 and 4 weeks. To elucidate this, we used RNA-sequencing to profile the skeletal muscle transcriptomes including lncRNAs and mRNAs, in diabetic GK and control Wistar rats at the age of 3 and 4 weeks. In total, there were 438 differentially expressed mRNAs (DEGs) and 401 differentially expressed lncRNAs (DELs) in skeletal muscle of 3-week-old GK rats compared with age-matched Wistar rats, and 1000 DEGs and 726 DELs between GK rats and Wistar rats at 4 weeks of age. The protein–protein interaction analysis of overlapping DEGs between 3 and 4 weeks, the correlation analysis of DELs and DEGs, as well as the prediction of target DEGs of DELs showed that these DEGs (*Pdk4*, *Stc2*, *Il15*, *Fbxw7* and *Ucp3*) might play key roles in hyperglycemia, glucose intolerance, and increased fatty acid oxidation. Considering the corresponding co-expressed DELs with high correlation coefficients or targeted DELs of these DEGs, our study indicated that these dysregulated lncRNA-mRNA pairs (NONRATG017315.2-*Pdk4*, NONRATG003318.2-*Stc2*, NONRATG011882.2-*Il15*, NONRATG013497.2-*Fbxw7*, MSTRG.1662-*Ucp3*) might be related to above biological processes in GK rats at the age of 3 and 4 weeks. Our study could provide more comprehensive knowledge of mRNAs and lncRNAs in skeletal muscle of GK rats at 3 and 4 weeks of age. And our study may provide deeper understanding of the underlying mechanism in T2D of GK rats at the age of 3 and 4 weeks.

# INTRODUCTION

It has been demonstrated that approximately 75% of the human genome is transcribed, and nearly 97% of genomic DNA cannot be translated into proteins (*Djebali et al., 2012*). These RNAs without protein-coding ability are known as non-coding RNAs (ncRNAs). Among them, long ncRNAs (lncRNAs) are more than 200 nucleotides in length (*Esteller, 2011*; *Guttman & Rinn, 2012*), exhibiting tissue-specific (*Cabili et al., 2011*; *Mercer et al., 2008*; *Tsoi et al., 2015*) and low expression levels (*Derrien et al., 2012*). They could promote

Corresponding authors
Shudai Lin, linsd@scut.edu.cn
Hongli Du, hldu@scut.edu.cn

(*Guil & Esteller, 2012*; *Luo & Chen, 2016*) or suppress (*Espinoza et al., 2004*; *Peterlin, Brogie & Price, 2012*; *Rinn et al., 2007*) the expression level of their target genes. Furthermore, it has been revealed that the expression of lncRNAs was dysregulated in many diseases, such as type 2 diabetes (T2D) (*Akerman et al., 2017*; *Liu et al., 2014*; *Reddy et al., 2014*).

As it was uncovered, lncRNAs were closely correlated to T2D. Upregulated expression of lncRNA Meg3 could contribute to insulin resistance in *ob/ob* mice liver (*Zhu et al., 2016*). The *db/db* mice islets showed significantly decreased expression of lncRNA Meg3, and the islet-specific knockdown of lncRNA Meg3 resulted in less insulin synthesis and secretion but larger scale of β cell apoptosis, consequently lead to impaired glucose tolerance (*You et al., 2016*). The islet-specific lncRNA Tug1 knockdown mice exhibited an increased apoptosis ratio and a lower insulin secretion in the β cells (*Yin et al., 2015*; *You et al., 2016*). Besides, the reduced expression of lncRNA H19 could impair insulin sensitivity and decrease glucose uptake in muscle cells (*Gao et al., 2014*). Moreover, significantly decreased expression of H19 was observed in muscle of T2D patients (*Gao et al., 2014*), suggesting the importance of skeletal muscle lncRNAs to the development of T2D. As one of the target tissues of insulin, skeletal muscle is burdened with 70%–80% postprandial glucose disposal responsibility (*Baron et al., 1988*; *DeFronzo et al., 1981*). Therefore, lncRNAs in skeletal muscle might play critical roles in regulating whole-body glucose homeostasis and T2D development.

As a non-obese model for T2D, Goto-Kakizaki (GK) rats are produced by selective breeding from Wistar rats with impaired glucose tolerance (*Goto, Kakizaki & Masaki, 1976*; *Kitahara et al., 1978*). GK rats show postprandial glucose intolerance and insulin resistance in skeletal muscle and adipose tissue (*Bisbis et al., 1993*; *Portha et al., 2012*), and exhibit hyperglycemia during age of 3–4 weeks (*Ando et al., 2018*). Though GK rats has been found to exhibit defects in skeletal muscle and their related mRNA expression level has been investigated (*Dadke et al., 2000*; *Steiler et al., 2003*), the regulation mechanism of skeletal muscle lncRNA to postprandial hyperglycemia in GK rats at the age of 3 and 4 weeks is still indistinct.

To explore the role of skeletal muscle lncRNAs in hyperglycemia development, we compared the skeletal muscle transcriptomes between T2D GK rats and control Wistar rats, to find out the differentially expressed mRNAs (DEGs) and differentially expressed lncRNAs (DELs). Subsequently, we conducted protein–protein interaction analysis, screened the co-expressed lncRNA-mRNA pairs with high correlation coefficients, and predicted the target mRNAs of DELs and the target microRNAs (miRNAs) of key DEGs and DELs. Our results suggested that the dysregulated lncRNAs might be implicated in hyperglycemia, glucose intolerance, as well as dysregulated glucose and fatty acid oxidation in skeletal muscle of GK rats at the age of 3 and 4 weeks. These findings might help us understand more about the regulation mechanism of skeletal muscle lncRNAs in T2D development.

## MATERIAL AND METHODS

### Ethical approval

The study was approved by the institutional review board of the Guangdong Key Laboratory of Laboratory Animals. All protocols were carried out in accordance with the guidelines

of the Institutional Animal Care and Use Committee (IACUC) (Ethics certificate No.: IACUC2014029).

## Animal breeding and tissues samples collection

Four groups of rats (diabetic male GK rats and control male Wistar rats at 3 weeks of age, diabetic male GK rats and diabetic male GK rats at 4 weeks of age, $n = 10$ each group), totally 40 subjects were used in this study. Rats were raised in a room with 12 h dark: 12 h light cycle, 20 to 25 °C temperature and $60 \pm 5\%$ humidity, at the SLAC Laboratory Animal Co., Ltd. (Shanghai, China) (*Almon et al., 2012*; *Nie et al., 2017*; *Nie et al., 2011*; *Xue et al., 2011*). All animals were free access to food and water. Body weight of each rat was measured weekly by weighing. Food disappearance was measured by weighing the difference in the weight of feed added and the feed remaining. The behavior of rats including feeding, drinking, sleeping and digging were observed. Blood samples were collected from the orbital plexus veins behind the eyeball using EDTA (4 mM final concentration) as an anticoagulant. Plasma was obtained from blood after centrifugation (2000×g, 4 °C, 15 min), divided into aliquots, and then stored at −80 °C. All rats were administered anesthesia with pentobarbital sodium (intraperitoneal, 50 mg/kg body weight), then were killed by cervical dislocation. Samples of gastrocnemius muscle of each rat were harvested, followed by rapidly frozen in liquid nitrogen, and stored at −80 °C for future studies (*Nie et al., 2017*; *Nie et al., 2011*). Six gastrocnemius muscle samples from six rats each group were selected randomly for RNA-sequencing in the present study.

## Measurement of plasma glucose and insulin concentration

The automatic Dry Biochemical Analyzer FUJIFILM DRI-CHEM 7000i with GLU-PIII slides (Fujifilm, Saitama, Japan) was used to measure random plasma glucose concentration. And Thermo scientific Rat Insulin ELISA Kit (Cat#ERINS, Invitrogen, Waltham, MA, USA) was used to measured plasma insulin concentration. Assays were conducted according to the manufacturer's instructions.

## RNA extraction and sequencing

Total RNA for RNA-sequencing was extracted from red part of each gastrocnemius muscle using TRIzol Reagent (Cat#15596-018, Life Technologies, Carlsbad, CA, USA) following the manufacturer's instructions. RNA integrity and concentration were measured by the Bioanalyzer 2100 system (Agilent Technologies, Santa Clara, CA, USA). Ribosomal RNA was removed using Epicentre Ribo-Zero$^{TM}$ Gold Kits (Epicentre, Madison, WI, USA) according to the manufacturer's instructions. RNA-sequencing was performed on Illumina HiSeq X Ten system (Illumina) following the HiSeq X Ten User Guide to generate 150 bp paired-end reads.

## Analysis of differentially expressed mRNAs and lncRNAs

After quality control and filtering of low quality reads, we used STAR (*Dobin & Gingeras, 2015*) version 020201 to align the cleaned reads of each sample to the Rattus norvegicus reference genome (Ensembl Rnor_6.0 version 92) with the parameters of –quantMode GeneCounts –outSAMstrandField intronMotif –outSAMtype BAM

SortedByCoordinate –outSAMtype BAM SortedByCoordinate –twopassMode Basic. All the corresponding annotation files of Rattus_norvegicus.Rnor_6.0.92.gtf (https://doi.org/10.6084/m9.figshare.11786277) and NONCODEv5_rat_rn6_lncRNA.gtf were downloaded from the Ensembl database54 and NONCODE version v5.0 (http://www.noncode.org/datadownload/NONCODEv5_rat_rn6_lncRNA.gtf.gz), respectively. Cufflinks were used for alignment of novel transcripts. Then the coding-probability of novel transcripts were identified by CPC2 (*Kang et al., 2017*), CPAT (*Wang et al., 2013b*) and CNCI (*Sun et al., 2013*). The novel transcripts with low coding-probability, or without coding-probability should meet the criteria: coding_probability score less than 0.5 in CPC2 and CPAT, and identified as noncoding by CNCI. Those novel transcripts meet criteria above with ≥ 200 bp in length and at least two exons, were defined as novel lncRNAs. Stringtie (*Pertea et al., 2015*) version 1.3.0 was used to assemble novel lncRNAs, annotated lncRNAs and annotated mRNAs transcripts. The novel lncRNAs were shown in Table S1. Ballgown R package (*Frazee et al., 2015*) version 2.10.0 was used to estimate the fragments per kilobase of exon per million fragments mapped (FPKM) of lncRNAs and mRNAs. The lncRNAs and mRNAs were filtered with FPKM < 0.5 (*Moran et al., 2012*). The FPKM from four groups of rats correspond to normal distribution based on the shapiro.test of Shapiro–Wilk test. The normal distribution of FPKM of four groups of rats were shown in Table S2. Next, the FPKM of GK rats at the age of 3 weeks were compared to Wistar rats at the age of 3 weeks, and GK rats at the age of 4 weeks were compared to Wistar rats at the age of 4 weeks. Thus, the differentially expressed lncRNAs (DELs) and differentially expressed mRNAs (DEGs) were obtained by Bayes-regularized $t$-test with an false discovery rate (FDR) correction using Cyber-T bayesreg (*Kayala & Baldi, 2012*). FDR < 0.05 was regarded as statistically significant. The power of test was calculated by pwr.t.test in R package pwr. The flowchart of data analysis was shown in Fig. 1.

## Analysis of KEGG pathways and GO

A Database for Annotation, Visualization and Integrated Discovery (DAVID) version 6.8 was used to obtain the Kyoto Encyclopedia of Genes and Genomes (KEGG) pathways and biological process in Gene Ontology (GO). The statistical significance threshold was $P < 0.05$.

## Protein–protein interaction

An online database resource Search Tool for the Retrieval of Interacting Genes (STRING) (*Szklarczyk et al., 2011*) version 11.0 was performed to analyze protein–protein interaction of overlapping upregulated mRNAs and downregulated mRNAs between 3 and 4 weeks, respectively. After filtering disconnected nodes, we selected the minimum confidence score above 0.4 of the interaction. The confidence score was a combined score of neighborhood on chromosome, gene fusion, phylogenetic cooccurrence, homology, co-expression, experimentally determined interaction, database annotated, and automated text-mining. Those connected nodes with confidence score were downloaded for constructing the networks of protein–protein interaction.

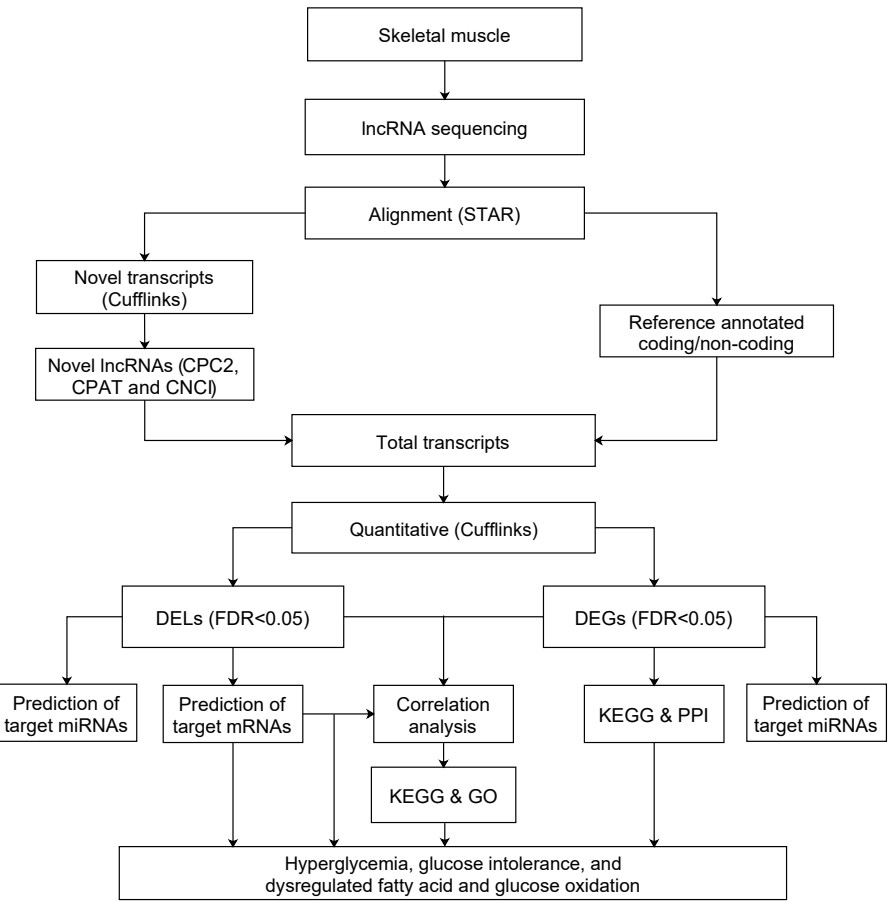

**Figure 1** **Flowchart of data analysis pipeline.** FDR, false discovery rate. DEGs, differentially expressed mRNAs; DELs, differentially expressed lncRNAs; PPI, protein–protein interaction. GO, Gene Ontology; KEGG, Kyoto Encyclopedia of Genes and Genomes.

## The correlation analysis of lncRNAs and mRNAs

Python version 3.6.4 was conducted to calculate the Pearson correlation coefficients of lncRNAs and mRNAs. Those selected co-expressed lncRNA-mRNA pairs met the following criteria: correlation coefficient value > 0.9, and the absolute fold change of these DEGs and DELs ≥ 1.5. Then, pwr.r.test in R package pwr was carried out to calculate the power of the correlation.

## Prediction of target genes of DELs

Since over 65% of lncRNAs were located within 10 kb of genes (*Jia et al., 2010*), we utilized the University of California Santa Cruz (UCSC) genome browser to identify potential cis-target genes located within 10 kb of lncRNAs (*Liang et al., 2017*). Then, the Basic Local Alignment Search Tool (BLAST) was applied to screen mRNAs that have complementary sequences to lncRNAs, followed by RNAplex (*Liang et al., 2017*) to identify trans-regulated target genes of lncRNAs. Subsequently, DELs and their corresponding target DEGs were obtained.
## Prediction of microRNAs both targeted to key DELs and DEGs

We predicted the target microRNAs (miRNAs) of DELs and DEGs in key lncRNA-mRNA pairs. TargetScan (*Agarwal et al., 2015*) was carried out to predict the target miRNAs of key DEGs, and the RNAhybrid (*Rehmsmeier et al., 2004*) was applied to predict the target miRNAs of key DELs.

## Construction of interaction network

Cytoscape version 3.6.1 was exerted to construct the networks of lncRNA-mRNA pairs and protein–protein interaction networks with those downloaded files including connected nodes with confidence score.

## Statistical analysis

All data were expressed as mean ± standard deviation (SD) unless otherwise noted. The significant difference was measured using a two-tailed student $t$-test. $P < 0.05$ was considered statistically significant.

# RESULTS

## The characteristics of rats

The plasma glucose concentration of GK rats was significantly higher than that of control Wistar rats at 3 and 4 weeks of age ($P < 0.001$, Table 1), which was in accordance with previous research (*Ando et al., 2018*). Besides, the plasma glucose concentration of 3-week-old GK rats was significantly higher than that of 4-week-old GK rats ($P < 0.001$, Table 1).

## Differentially expressed lncRNAs and mRNAs

In total, we got 438 and 1,000 differentially expressed mRNAs (DEGs) between GK and Wistar rats at 3 and 4 weeks of age, respectively (false discovery rate, FDR < 0.05) (Fig. 2A, Table S3). There were 401 and 746 differentially expressed lncRNAs (DELs) in GK rats compared with Wistar rats at 3 and 4 weeks of age, respectively (FDR < 0.05) (Fig. 2B, Table S4). Among the DEGs, 141 overlapping upregulated mRNAs and 103 overlapping downregulated mRNAs were detected between 3 and 4 weeks (Fig. 2A). A total of 91 overlapping upregulated lncRNAs and 124 overlapping downregulated lncRNAs were found between 3 and 4 weeks (Fig. 2B). From the results of enrichment pathway analysis of DEGs at 3 and 4 weeks, we found the insulin resistance pathway was the only one overlapping pathway between 3 and 4 weeks among top 10 KEGG pathways (Figs. 2C and 2D). The DEGs enriched in insulin resistance pathway were sterol regulatory element binding transcription factor 1 (*Srebf1*, also known as *Srebp1c*), solute carrier family 27 member 1 (*Slc27a1*), protein kinase C, theta (*Prkcq*), cAMP responsive element binding protein 3-like 1 (*Creb3l1*), forkhead box O1 (*Foxo1*), TBC1 domain family, member 4 (*Tbc1d4*, also termed as *AS160*), and carnitine palmitoyltransferase 1A (*Cpt1a*).

To obtain the interaction of the proteins encoded by 141 overlapping upregulated genes and 103 overlapping downregulated genes between 3 and 4 weeks, we analyzed protein–protein interaction of these proteins using STRING. Next, we constructed the

**Table 1** The characteristics of rats.

| Rats | Age, weeks | Plasma glucose, mmol/L | Plasma insulin, pmol/L | Weight, g |
|---|---|---|---|---|
| Wistar | 3 | $5.05 \pm 0.39$ | $157.66 \pm 64.52$ | $88.92 \pm 7.08$ |
|  | 4 | $6.65 \pm 0.49$ | $168.97 \pm 36.37$ | $139.42 \pm 9.62$ |
| GK | 3 | $7.82 \pm 0.43$*** | $197.83 \pm 57.69$ | $85.03 \pm 14.36$ |
|  | 4 | $11.13 \pm 0.29$*** | $147.16 \pm 59.92$ | $131.94 \pm 16.56$ |

**Notes.**

Values are means $\pm$ SD.

***$P < 0.001$ vs age-matched Wistar group $n = 10$.

SD, standard deviation

protein–protein interaction network (Figs. 3A and 3B). Then, the top 10 mRNAs according to the node degree among network and their corresponding node degrees were listed in Table S5 . The network among top 10 upregulated and downregulated node mRNAs were shown in Figs. 3C and 3D. The dysregulated genes *Srebf1*, *Slc27a1*, *Foxo1* and *Cpt1a* that enriched in insulin resistance pathway also existed in the network of top 10 upregulated and downregulated node mRNAs (Figs. 3C and 3D), which indicating that these four genes might be important for the development of hyperglycemia and T2D in GK rats at the age of 3 and 4 weeks.

## The co-expressed lncRNA-mRNA pairs with high correlation coefficients

To investigate the potential function of these DELs, we performed lncRNA-mRNA co-expression network analysis. After filtering, a total of 901 co-expressed lncRNA-mRNA pairs with high correlation coefficients were selected, including 136 DEGs and 120 DELs (Table S6). 136 DEGs were enriched in two KEGG pathways ($P < 0.05$), including transcriptional misregulation in cancer and pathways in cancer. But both pathways were not related to T2D. 136 DEGs were enriched in biological processes (Fig. 4A). Among these DEGs, 2 DEGs (pyruvate dehydrogenase kinase 4, *Pdk4* and *Cpt1a*) were enriched in "regulation of fatty acid oxidation", and 3 DEGs (*Foxo1*, *Pdk4* and SH2B adaptor protein 2, *Sh2b2*) were enriched in "insulin receptor signaling pathway". The top 10 nodes ranked by degrees in co-expressed lncRNA-mRNA network were consist of 7 mRNAs and 3 lncRNAs (Table 2). The network of these dysregulated mRNAs (*Cep19*, *Cpt1a*, *Ephx2*, *Foxo1*, *Pdk4*, *Sh2b2* and *Stc2*) and their co-expressed lncRNAs were shown in Fig. 4B. And the expression of key co-expressed lncRNA-mRNA pairs were shown in Figs. 4C and 4D. Notably, the dysregulated genes *Cpt1a*, *Foxo1* and *Pdk4* also appeared in the network of top 10 upregulated and downregulated mRNAs (Figs. 4C and 4D), indicating that these mRNAs might associate with hyperglycemia and T2D in GK rats at the age of 3 and 4 weeks.

However, other genes involved in fatty acid transport and β-oxidation were not significantly dysregulated in GK rats at the age of 3 and 4 weeks (Table S7). In our study, genes related to glycolysis and glycogen synthesis were not dysregulated in GK rats at 3 and 4 weeks of age (Table S7).

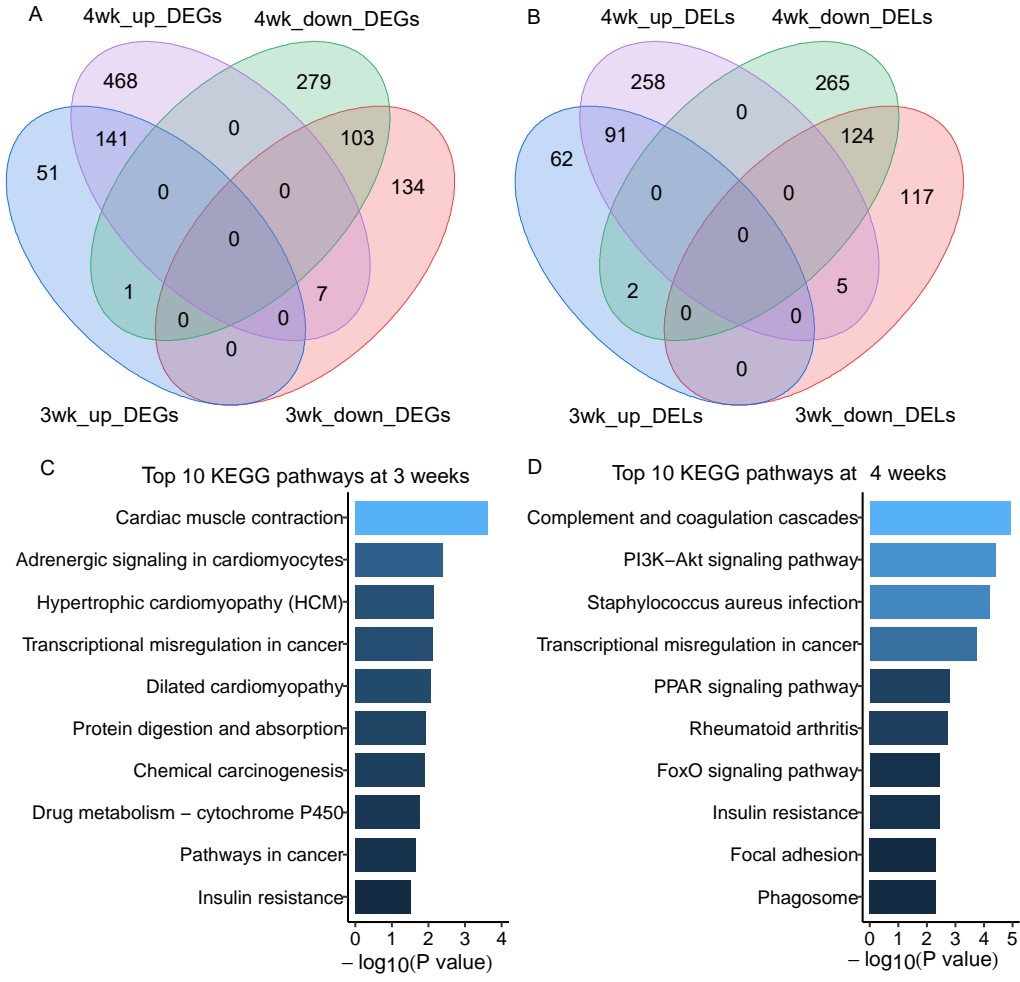

**Figure 2** **The differentially expressed lncRNAs and mRNAs in skeletal muscle of GK rats compared with aged-matched Wistar rats.** (A) Venn diagram of differentially expressed mRNAs (DEGs) in skeletal muscle of GK rats compared with aged-matched Wistar rats at the age of 3 and 4 weeks. (B) Venn diagram of differentially expressed lncRNAs (DELs) in skeletal muscle of GK rats compared with aged-matched Wistar rats at the age of 3 and 4 weeks. (C) The top 10 KEGG pathways of DEGs in GK rats compared with aged-matched Wistar rats at the age of 3 weeks. (D) The top 10 KEGG pathways of DEGs in GK rats compared with aged-matched Wistar rats at the age of 4 weeks. The 3wk_up_DEGs represents upregulated mRNAs at 3 weeks. The 4wk_up_DEGs represents upregulated mRNAs at 4 weeks. The 3wk_down_DEGs represents downregulated mRNAs at 3 weeks. The 4wk_down_DEGs represents downregulated mRNAs at 4 weeks. The 3wk_up_DELs represents upregulated lncRNAs at 3 weeks. The 4wk_up_DELs represents upregulated lncRNAs at 4 weeks. The 3wk_down_DELs represents downregulated lncRNAs at 3 weeks. The 4wk_down_DELs represents downregulated lncRNAs at 4 weeks.

## The predicated target mRNAs of differentially expressed lncRNAs

To identify the potential role of dysregulated lncRNAs in the development of hyperglycemia and T2D in GK rats at the age of 3 and 4 weeks, we predicted their cis- and trans-target mRNAs. A total of 15 predicted cis-target DEGs and 88 predicted trans-target DEGs were obtained in DELs at 3 weeks (Table S8). There were 31 predicted cis-target DEGs and 382 predicted trans-target DEGs in DELs at 4 weeks (Table S9). There were 32 overlapping

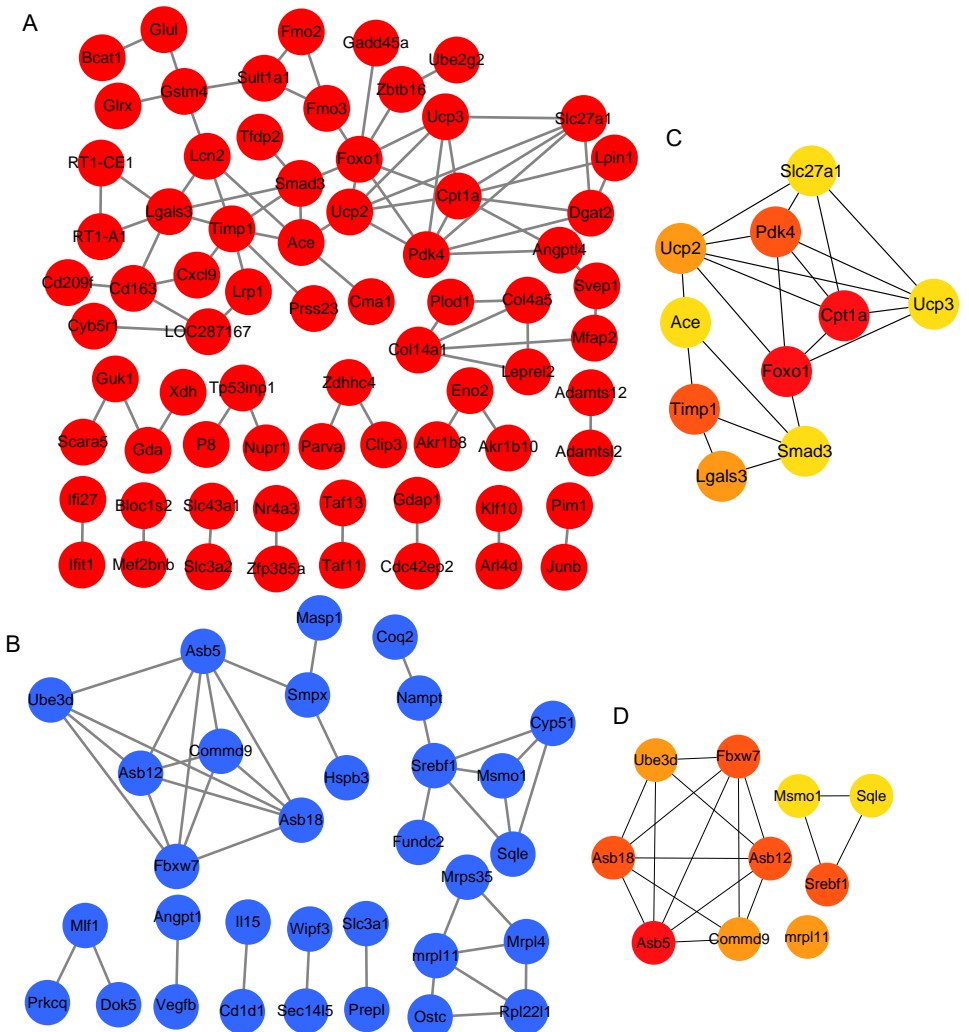

**Figure 3 The top 10 mRNAs identified in protein–protein interaction networks.** (A) The protein–protein interaction network of overlapping upregulated DEGs at 3 and 4 weeks. The red represents significantly upregulated mRNAs in GK rats at the age of 3 and 4 weeks. (B) The protein–protein interaction network of overlapping downregulated DEGs at 3 and 4 weeks. The blue represents significantly downregulated mRNAs in GK rats at the age of 3 and 4 weeks. (C) The top 10 upregulated mRNAs ranked by node degree. The darker of the color indicates the higher of connectivity degree. (D) The top 10 downregulated mRNAs ranked by node degree. The darker of the color indicates the higher of connectivity degree.

DEL-target DEGs between 3 and 4 weeks (Fig. 5A, Table S10). Network analysis for these overlapping DEGs found out 32 lncRNA-mRNA pairs, including 18 DEGs and 19 DELs (Fig. 5B). Among the 18 DEGs, interleukin 15 (*Il15*), F-box and WD repeat domain containing 7 (*Fbxw7*) and uncoupling protein 3 (*Ucp3*) were related to increased glycaemia (*Gray & Kamolrat, 2011*; *Zhao et al., 2018*), glucose intolerance (*Fujimoto et al., 2019*), and increased fatty acid oxidation (*Bezaire et al., 2005*). Among those 32 lncRNA-target mRNA pairs, 5 lncRNA-target mRNA pairs (NONRATG014028.2-*Pim1*, NONRATG011882.2-*Il15*, NONRATG013497.2-*Fbxw7*, NONRATG011747.2-*Mrps35*,

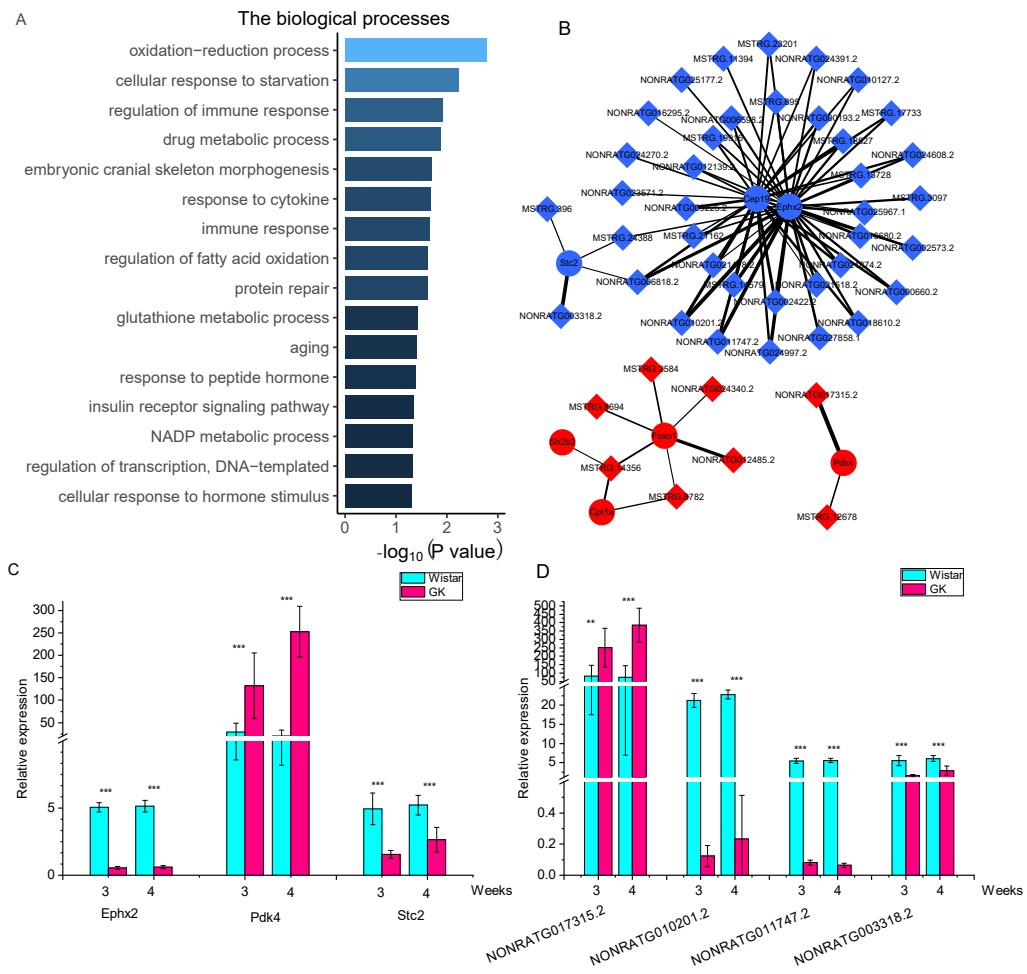

Figure 4 **The co-expressed lncRNAs-mRNAs pairs with high correlation coefficients.** (A) The biological process of dysregulated mRNAs in co-expressed lncRNAs-mRNAs pairs with high correlation coefficients. The GO terms of biological process were as follows : GO: 0055114, oxidation-reduction process; GO: 0009267, cellular response to starvation; GO: 0050776, regulation of immune response; GO: 0017144, drug metabolic process; GO: 0048701, embryonic cranial skeleton morphogenesis; GO: 0034097, response to cytokine; GO: 0006955, immune response; GO: 0046320, regulation of fatty acid oxidation; GO: 0030091, protein repair; GO: 0006749, glutathione metabolic process; GO: 0007568, aging; GO: 0043434, response to peptide hormone; GO 0008286, insulin receptor signaling pathway; GO: 0006739, NADP metabolic process; GO 0006355, regulation of transcription, DNA-templated; GO 0032870 cellular response to hormone stimulus. (B) The network of key genes and their co-expressed lncRNAs. The red represents upregulated gene in GK rats compared with aged-matched Wistar rats at the age of 3 and 4 weeks. The blue represents downregulated gene in GK rats compared with aged-matched Wistar rats at the age of 3 and 4 weeks. The diamond represents lncRNA, while the circle represents mRNA. The line between lncRNA and mRNA represents the co-expression coefficient. The range of correlation coefficients was from 0.9 to 0.993573997. (C). The relative expression of *Ephx2*, *Pdk4* and *Stc2*. (D) The relative expression of NONRATG017315.2, NONRATG010201.2, NONRATG011747.2, and NONRATG003318.2. Values are means ± SD, *$P < 0.05$, **$P < 0.01$, ***$P < 0.001$ vs age-matched Wistar group, $n = 6$.

**Table 2 Top 10 nodes ranked by the degree in co-expressed lncRNA-mRNA network.**

| ID (Name) | Type | Degree |
|---|---|---|
| ENSRNOG00000017286 (*Ephx2*) | mRNA | 34 |
| ENSRNOG00000005177 (*Tp53i3*) | mRNA | 33 |
| ENSRNOG00000026493 (*Cdnf*) | mRNA | 32 |
| MSTRG.2584 | lncRNA | 31 |
| ENSRNOG00000024924 (*Cep19*) | mRNA | 31 |
| ENSRNOG00000000473 (*Pfdn6*) | mRNA | 30 |
| MSTRG.8694 | lncRNA | 28 |
| MSTRG.14356 | lncRNA | 26 |
| ENSRNOG00000010802 (*Ube3d*) | mRNA | 25 |
| ENSRNOG00000016937 (*Mtfr1l*) | mRNA | 25 |

and MSTRG.1662-*Ucp3*) appeared in 901 co-expressed lncRNA-mRNA pairs with high correlation coefficients, suggesting these dysregulated lncRNA-mRNA pairs might involve in the hyperglycemia and T2D of GK rats at the age of 3 and 4 weeks. And the relative expression of NONRATG011882.2-*Il15*, NONRATG013497.2-*Fbxw7*, and MSTRG.1662-*Ucp3* pairs were shown in Fig. 5C.

**The target microRNAs (miRNAs) of key DEGs and DELs**

To explore the role of lncRNAs in the expression of mRNAs, we predicted the target miRNA of key DELs and DEGs. Then we obtained the overlapping miRNAs targeted both DEGs and DELs in key lncRNA-mRNA pairs (Fig. 6), which provided miRNAs linkers between DELs and DEGs. We found that rno-miR-139-5p, rno-miR-486 and rno-miR-93-5p target both MSTRG.14356 and *Foxo1* (Fig. 6). We got three target miRNAs rno-miR-20b-5p, rno-miR-27a-3p and rno-miR-17-5p of MSTRG.2584 and *Foxo1* (Fig. 6). The target miRNAs of *Stc2* and MSTRG.2584 were rno-miR-24-3p, rno-miR-532-5p, rno-miR-181a-5p and rno-miR-181b-5p (Fig. 6). MiRNAs rno-miR-195-5p, rno-miR-181b-5p, rno-miR-23b-3p, rno-miR-139-5p and rno-miR-23a-3p were the target miRNAs of MSTRG (Fig. 6).12678 and *Pdk4*. Three miRNAs including rno-miR-34a-5p, rno-miR-125a-5p and rno-miR-125b-5p targeted MSTRG.1662 and *Ucp3* (Fig. 6). Additionally, the target miRNA of NONRATG013497.2 and *Fbxw7* was rno-miR-24-3p (Fig. 6). And rno-miR-326-3p was the target of NONRATG011882.2 and *Il15* (Fig. 6).

## DISCUSSION

In the present study, we obtained mRNA and lncRNA expression profiles of skeletal muscle of GK and Wistar rats at 3 and 4 weeks of age by RNA-sequencing. In total, 438 DEGs and 401 DELs were obtained in skeletal muscle of GK rats compared with Wistar rats at the age of 3 weeks (FDR < 0.05), 1,000 DEGs and 746 DELs at 4 weeks of age (FDR < 0.05). To address the function of those DELs, we screened the co-expressed lncRNA-mRNA pairs with high correlation coefficients, predicted the target mRNAs of DELs and predicted miRNAs targeted both DEGs and DELs. In considering previous studies, our results indicated that the dysregulated expressed lncRNA-mRNA pairs might be implicated in

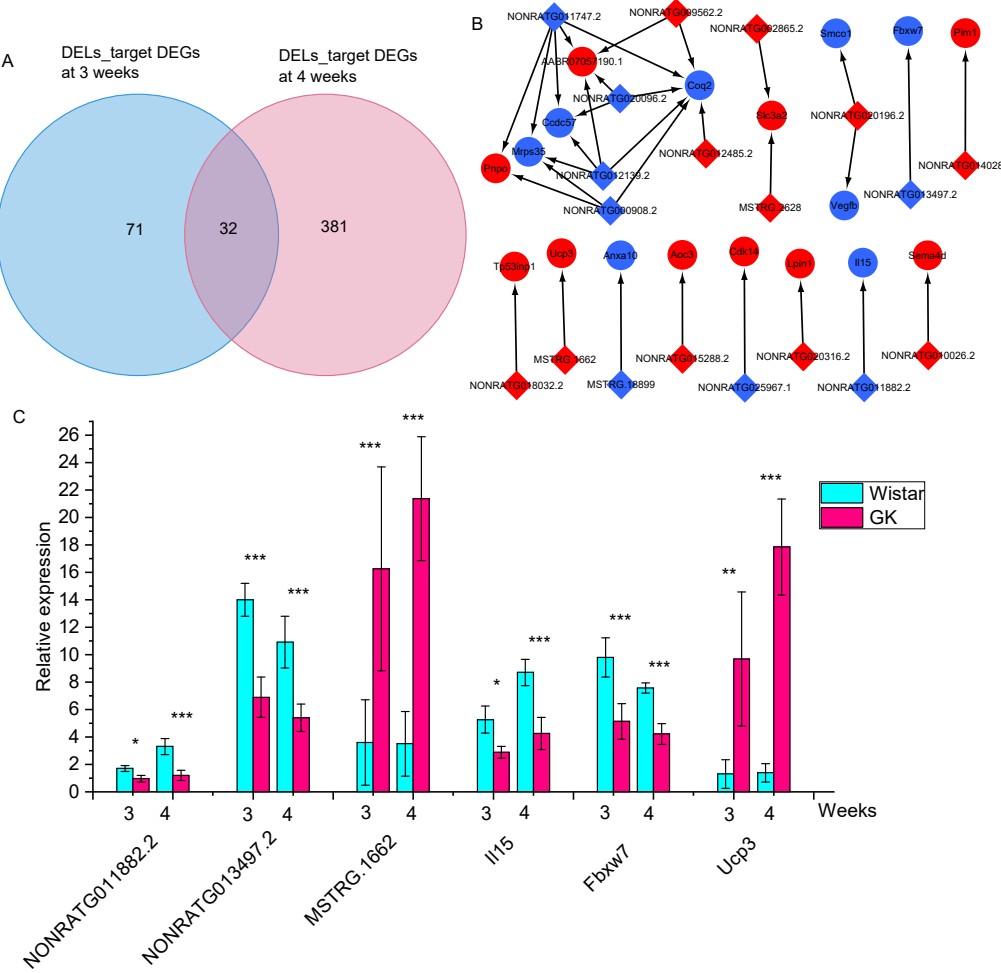

**Figure 5** **The differentially expressed target mRNAs for differentially expressed lncRNAs.** (A) Venn diagram of lncRNA-target mRNAs at 3 and 4 weeks. The DELs_target DEGs represents the target differentially expressed mRNAs of differentially expressed lncRNAs. (B) The network of overlapping lncRNA-target mRNAs at 3 and 4 weeks. The blue represents downregulated expression in GK rats compared with aged-matched Wistar rats at 3 and 4 weeks. The diamond represents lncRNA, the circle represents mRNA. (C) The relative expression of NONRATG011882.2-*Il15*, NONRATG013497.2-*Fbxw7*, and MSTRG.1662-*Ucp3*. Values are means ± SD, *$P < 0.05$, **$P < 0.01$, ***$P < 0.001$ vs age-matched Wistar group, $n = 6$. The red represents upregulated expression in GK rats compared with aged-matched Wistar rats at the age of 3 and 4 weeks.

hyperglycemia, glucose intolerance, and increased fatty acid oxidation in GK rats at the age of 3 and 4 weeks. However, the annotation of lncRNAs is incomplete, and the function of them has not been explained clearly. Thereby, further studies are necessary to reveal their potential function.

The DEGs *Slc27a1*, *Cpt1a*, *Srebf1*, and *Foxo1* were enriched in insulin resistance pathway and also appeared in the network of top 10 upregulated and downregulated mRNAs, indicating these four mRNAs might play key roles in the development of hyperglycemia and T2D in GK rats at the age of 3 and 4 weeks. It has been demonstrated that SLC27A1 was

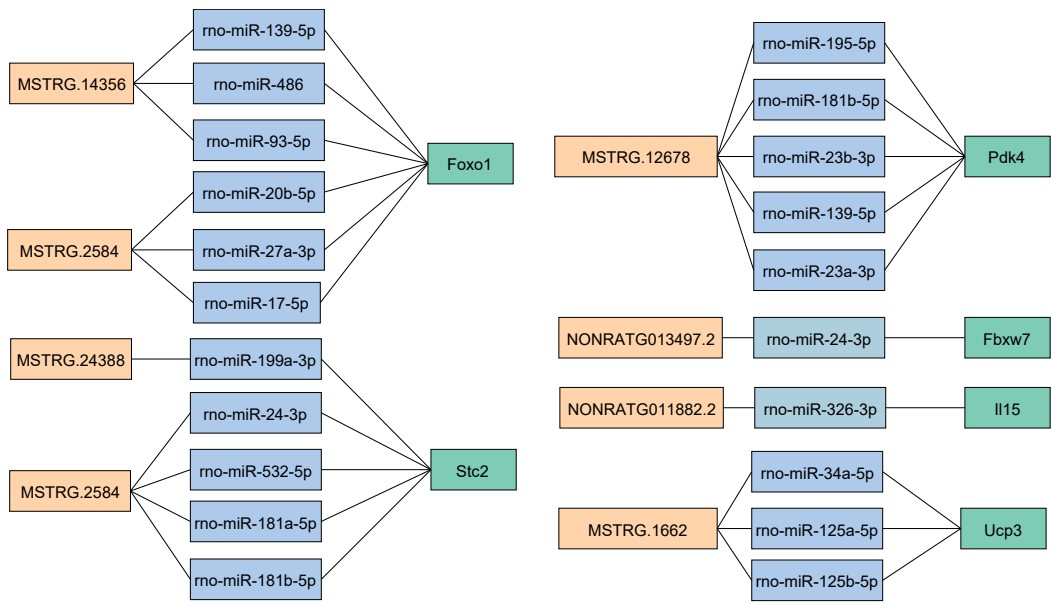

**Figure 6** **The target miRNAs of DEGs and DELs in key lncRNA-mRNA pairs.** The light orange represents lncRNA. The light blue represents miRNA. The light green represents mRNA.

implicated in the regulation of fatty acid transport and oxidation. Overexpression of *Slc27a1* could increase fatty acid uptake and oxidation in L6E9 skeletal muscle cells (*Sebastian et al., 2009*). CPT1A encoded by *Cpt1a* is responsible for transport long-chain fatty acid into mitochondria. And overexpression of *Cpt1a* could lead to enhanced fatty acid oxidation in hepatocytes, β-cells and muscle cells (*Akkaoui et al., 2009*; *Herrero et al., 2005*; *Perdomo et al., 2004*; *Stefanovic-Racic et al., 2008*). Srebf1 is a transcription factor regulating fatty acid synthesis. The *ob/ob* mice with inactivated SREBF1 showed reduced hepatic fatty acid synthesis (*Moon et al., 2012*). Blood glucose was significantly higher in $Srebf1^{-/-}$ mice than in $Srebf1^{+/+}$ mice (*Jang et al., 2016*). It has been unveiled that inhibiting expression of *Foxo1* could increase glucose oxidation in mouse heart (*Gopal et al., 2017*). Moreover, TBC1D4 was reported to be involved in glucose uptake. Whole-body knockout *Tbc1d4* mice exhibited markedly decreased insulin-stimulated glucose uptake in skeletal muscle (*Lansey et al., 2012*; *Wang et al., 2013a*). Therefore, in our study, the significantly increased expression of *Slc27a1*, *Cpt1a* and *Foxo1* might associate with increased fatty acid transport and oxidation in skeletal muscle of GK rats at 3 and 4 weeks of age. Randle et al. pointed out, increased oxidation of fatty acids could repress glucose oxidation (*Hue & Taegtmeyer, 2009*); (*Randle et al., 1963*). Thus, increased fatty acid transport and oxidation might be related to increased glucose concentration in GK rats at the age of 3 and 4 weeks. Additionally, the significantly reduced expression of *Tbc1d4* might be related to decreased glucose uptake in skeletal muscle of GK rats at 3 and 4 weeks of age. Taken together, our results indicated that *Srebf1*, *Slc27a1*, *Foxo1*, *Tbc1d4*, and *Cpt1a* might be related to increased glycaemia in GK rats at 3 and 4 weeks of age.

As GK rats are produced from Wistar rats with impaired glucose tolerance, which plays an essential role in T2D development of GK rats. In this study, *Ephx2*, *Stc2*, *Cep19*, *Il15* and *Fbxw7* genes were found to be associated with impaired glucose tolerance and hyperglycemia. Elevated epoxyeicosatrienoic acids, which was inhibited by the enzyme encoded by *Ephx2*, could improve insulin-stimulated glucose uptake in skeletal muscle of *db/db* mice (*Shim et al., 2014*). Additionally, previous studies showed that mice with whole-body knockout *Ephx2* exhibited improved insulin secretion (*Luo et al., 2010*; *Luria et al., 2011*). Moreover, plasma glucose clearance was faster in whole-body knockout *Ephx2* mice than that in wild-type mice (*Luria et al., 2011*). As the expression of *Ephx2* was lower in skeletal muscle than in kidney and liver, the reduced *Ephx2* might have a weaker effect on glucose clearance in skeletal muscle of GK rats at the age of 3 and 4 weeks. Whole-body *Stc2* and *Cep19* knockout mice displayed significantly increased circulating glucose concentration (*Lopez et al., 2018*), and markedly impaired glucose tolerance and insulin resistant (*Shalata et al., 2013*), respectively. It has been explored that overexpressed *Il15* transgenic mice showed better glucose tolerance compared to wild-type mice, and Glut4 translocation was promoted in skeletal muscle by AMP-Activated protein kinase pathway (*Fujimoto et al., 2019*). In addition, skeletal muscle-specific overexpression of *Il15* transgenic mice displayed greater insulin sensitivity and decreased glucose concentration (*Quinn et al., 2011*). Liver-specific *Fbxw7* knockout mice presented hyperglycemia, glucose intolerance, and insulin resistance (*Zhao et al., 2018*). Thus, the significantly downregulated *Stc2*, *Il15*, and *Fbxw7* might associate with hyperglycemia and impaired glucose tolerance in GK rats at 3 and 4 weeks of age. Since targeted lncRNAs of these significantly downregulated mRNAs had a high correlation coefficient, the co-expressed pairs lncRNA-mRNA pairs, such as NONRATG003318.2-*Stc2*, NONRATG011882.2-*Il15*, and NONRATG013497.2-*Fbxw7* be related to hyperglycemia and impaired glucose tolerance in GK rats at the age of 3 and 4 weeks.

It is well known that increased fatty acid oxidation could inhibit glucose oxidation in heart and skeletal muscle (*Hue & Taegtmeyer, 2009*; *Randle et al., 1963*). Thus, the dysregulated fatty acid oxidation might affect circulating glucose concentration. *Pdk4* and *Ucp3* were found to be associated with increased fatty acid oxidation. Upregulated *Pdk4* could decrease glucose oxidation and enhance fatty acid oxidation in myocardium and skeletal muscle (*Sugden & Holness, 2003*; *Zhao et al., 2008*). As one transcriptional factor, Foxo1 could regulate the expression of *Pdk4*, and the inhibition of it could increase glucose oxidation in mouse heart (*Gopal et al., 2017*). Ucp3, located in mitochondrial inner membrane, expressed predominantly of skeletal muscle in humans and rodents (*Boss et al., 1997*). Whole-body *Ucp3* overexpression mice showed increased activity of enzymes that implicated in fatty acid oxidation in skeletal muscle (*Bezaire et al., 2005*). The significantly decreased rate of long-chain fatty acid oxidation was observed in rat heart with partial loss of *Ucp3* gene ($Ucp3^{+/-}$) (*Edwards et al., 2018*). Hence, in our study, the significantly increased *Pdk4*, and *Ucp3* and their corresponding co-expressed or targeted lncRNAs, including NONRATG017315.2-*Pdk4*, and MSTRG.1662-*Ucp3* might contribute to increased glycaemia and increased fatty acid oxidation in GK rats at 3 and 4 weeks of age.

Glycolysis and glycogen synthesis were demonstrated to be involved in glucose homeostasis (*Hwang et al., 1995*; *Rothman, Shulman & Shulman, 1992*; *Shulman et al., 1990*). However, in our study, genes related to glycolysis and glycogen synthase were not dysregulated in GK rats at the age of 3 and 4 weeks.

Recently, evidences showed that lncRNAs could regulate mRNAs by interacting with microRNAs (miRNAs) (*Zhang & Zhu, 2014*). Among the target miRNA of mRNAs we predicted, miR-139 has been identified could target *Foxo1* directly and inhibit its expression in mice hepatocytes (*Hasseine et al., 2009*). What's more, miR-139 overexpression leads to markedly reduced *Foxo1* level, and the inhibition of miR-139 contribute to increased *Foxo1* level (*Yan et al., 2018*). Moreover, *Foxo1* was the target gene of miR-486-5p (*Liu et al., 2019*). Thus, the DELs in key lncRNA-mRNA pairs might regulate mRNAs level through binding to their common miRNAs. Further studies are needed to identify the target miRNAs of DEGs and DELs and measure the miRNAs profiles in skeletal muscle of GK rats in the future.

## CONCLUSIONS

In the present study, we found that the dysregulated lncARNA-mRNA pairs (NONRATG017315.2-*Pdk4*, NONRATG003318.2-*Stc2*, NONRATG011882.2-*Il15*, NONRATG013497.2-*Fbxw7* and MSTRG.1662-*Ucp3*) might be implicated in hyperglycemia, glucose intolerance, as well as dysregulated glucose and fatty acid oxidation in GK rats at 3 and 4 weeks of age. These results may provide more comprehensive knowledge about mRNAs and lncRNAs in skeletal muscle of GK rats at the age of 3 and 4 weeks. Furthermore, these results may serve as important resources for future studies to investigate the regulatory mechanism of lncRNAs in skeletal muscle of GK rats at the age of 3 and 4 weeks.

### Funding
This work was supported by the National Key R&D Program of China (2018YFC0910201), the Key R&D Program of Guangdong Province (2019B020226001), and the Science and Technology Planning Project of Guangzhou (201704020176). The funders had no role in study design, data collection and analysis, decision to publish, or preparation of the manuscript.

### Grant Disclosures
The following grant information was disclosed by the authors:
National Key R&D Program of China: 2018YFC0910201.
Key R&D Program of Guangdong Province: 2019B020226001.
Science and Technology Planning Project of Guangzhou: 201704020176.

### Competing Interests
The authors declare there are no competing interests.

## Author Contributions

- Wenlu Zhang conceived and designed the experiments, performed the experiments, analyzed the data, prepared figures and/or tables, authored or reviewed drafts of the paper, and approved the final draft.
- Yunmeng Bai and Zixi Chen analyzed the data, prepared figures and/or tables, authored or reviewed drafts of the paper, and approved the final draft.
- Xingsong Li performed the experiments, analyzed the data, prepared figures and/or tables, and approved the final draft.
- Shuying Fu, Lizhen Huang and Shudai Lin performed the experiments, authored or reviewed drafts of the paper, and approved the final draft.
- Hongli Du conceived and designed the experiments, authored or reviewed drafts of the paper, and approved the final draft.

## Animal Ethics

The following information was supplied relating to ethical approvals (i.e., approving body and any reference numbers):

The institutional review board of the Guangdong Key Laboratory of Laboratory Animal approved this study (IACUC2014029).

## Data Availability

Raw sequencing reads are available at NCBI SRA: PRJNA483006.

## Supplemental Information

Supplemental information for this article can be found online at http://dx.doi.org/10.7717/peerj.8548#supplemental-information.

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
