# Peer review of "Comprehensive analysis of long non-coding RNAs and mRNAs in skeletal muscle of diabetic Goto-Kakizaki rats during the early stage of type 2 diabetes"

_PeerJ, doi:10.7717/peerj.8548_

## Round 0.1 · original submission · Minor Revisions

Dear authors,

Please answer the questions of the reviewers and perform appropriate editing of your text.

Reviewer 1 ·

Basic reporting

The authors provide interesting data about the changes in expression of long non-coding RNAs in Goto-Kakizaki rats at the age of 3-4 weeks compared with non-diabetic Wistar rats.
English language needs editing. Often, the sentences are not formed correctly, which makes the understanding difficult.
References are cited appropriately, reference list is correct.Manuscript is structured according to PeerJ standards. Abstract corresponds to PeerJ requirements, but language and style editing is needed.
Figures are relevant, high quality. Figures are well labelled & described, however, Fugure 4 A requires more comprehensive designation of the biological processes to my opinion. For example, „response to cytokine”, „response to peptide hormone”, „glutathion metabolic pathway”, „NADP” metabolic process”, „cellular response to hormone stimulus” are too simplistic and general terms - either more correct terms or explanation in the legend is needed.
Raw data are supplied

Experimental design

Positive: the manuscript corresponds to the Scope of PeerJ, and refers to basic diabetology area. Research question is well defined, relevant & meaningful. It is stated how the research fills an identified knowledge gap. Rigorous investigation performed to high technical.
Weaknesses:
To my opinion, description of methods needs refining. Importantly, it is not clear how the samples of gastrocnemius muscle were obtained – were the rats euthanized or anaesthetized? Was the dissection of the tissue performed under microscope and miocytes were separated or whole tissue including blood vessels and connective tissue was used? The authors should either give the details or provide a reference, if the procedure has been described elsewhere.
Line 41-42 – The paper of Maeda et al refers to mouse transcriptome, therefore it is not correct to put it into context after mentioning human transcriptome (decribed by Djebali et al).
Line 67 – add „tissue” after „adipose”
Line 68 and thoughout the text and abstract – please use „age of 3-4 weeks”, not „stage”.
Line 71 – „the mechanism of hyperglycemia by which skeletal muscle lncRNAs
regulated in the early stage of T2D in GK rat is still indistinct”- these sentence is not clear to me. Wjat is the cause and what is the consequence? Does lncRNAs regulate the mechanism of hyperglycamia? Please rewrite the sentence clearly
line 79-80 – again, I think the age of GKZ rats of 3-4 weeks is not the same as „early stage” diabetes. In general, diabetes has no stage. It is either present (confirmed by hyperglycaemia), or not. The term „glucose intolerance” or „prediabetes” might be used for glucose values higher than normal but below the diabetic range – please referē to animal reference values to define the condition of the animals at the age of 3-4 weeks.

Line 89 – instead of “Four group rats” please write “four groups of rats”
Line 92-93 – „orbital plexus veins”
Line 95 – please write „samples of m.gastrocnemius of each rat” instead of „Ten gastrocnemius samples” „six gastrocnemius samples”
Line 100 – please provide kit number and manufacturer (Invitrogen?)
Line 104 – use „gastrocnemius muscle” instead of „gastrocnemius”
Line 116-118 – please rewrite this sentence in a more comprehensive style
Line 127, 131, 135 – please write the full name of DAVID, STRING, Python, Target Scan, RNAhybrid, Cytoscape analysis tools/programmes
line 151-153 – Description of statistical analysis is too short and general. If 4 groups of rats were compared, the authors should have used Anova or Kruskal -Wallis test before pair-wise comparisons. Was the normality of data tested? t-test can only be used for normally distributed data. If the data do not correspond to normal distrubution, they should be presented as medians with corresponding interquartile range or in another appropriate way (not means)
Line 162-163 “The plasma glucose concentration of GK rats aged 3 weeks was
163 significantly increased than that of GK rats aged 4 weeks” – maybe “significantly higher than” sounds better
Line 168 – FDR – abbreviation is used for the first time and is not explained
Line 195 – „processes” instead of „process” should be used
201- „Intolerance” instead of „intolerant”
211 I would suggest „in GK rats at the age of 3 and 4 weeks” instead of „and T2D at 3 and 4 weeks”
213-214 reference is needed after „Glycolysis and glycogen synthase were demonstrated to be involved in glucose homeostasis”. This sentence might be more appropriate in Discussion section
242 – please use „RNA sequencing” instead of „RNA-seq”
322 –„ miR-139 has been identified could target Foxo1 directly and inhibit Foxo1 expression 323 in mice hepatocytes (Hasseine et al. 2009)” – the sentence contains unnecessary words.
323- “leads”
327-329 – why past is used in this sentence? I think you should use the present form of the verbs.
335-338 The last sentence should be rewritten in a comprehensive way, it is hard to understand what did the authors want to say.

Validity of the findings

Positive- The authors have demonstrated sufficient data to justify the need for this study. Negative results have been demonstrated and discussed. All underlying data have been provided.
Negative - The choice of the methods for statistical analysis needs to be explained thoroughly in the methods, or, recalculation of the data using more appropriate methods is recommended.
Conclusions are not well stated (mot probably due to linguistic problems)

·

Basic reporting

I commend the authors for the throughout RNA seq data analysis using various methods and providing maximum information from the data. In addition the results are well framed within existing knowledge.
However two main issues can be highlighted from the manuscript.
First the wellbeing of experiment animals which could have significant impact on results and second about the way how measurements are reported. Also information regarding settings in bioinformatical analyses could be provided in more detail. English proofreading by native speaker is advised.


A) English proofreading is advised. For example: line 67: I suppose "tissue" is missing (or "adipocytes"), line 89: "Four groups of...", ect..

B) Figure 3 legend. There is confusing information in legend part 3C " The blue represents significantly downregulated mRNAs in GK rats at 3 and 4 weeks of age.": But there are no blue colored nodes in figure 3C.
Also part 3D "The color represents the degree." does not explain HOW figure 3D is represented by color.

C) Figure 4 legend and Figure 5 legend: "Values are means ± SEM, *P < 0.001, **P < 0.001, ***P < 0.001 vs age-matched Wistar". All star levels display same significance, must be corrected.
For improved description of results bar graphs should show SD instead

D) Table 1. Please provide SD. Are five decimal places in reporting plasma insulin necessary? I suggest reducing to relevant level of precision.

Experimental design

A) Please elaborate on animal housing (lines 91-92) whether the rats were in social contact with each other.
What was the evidence for the need to house rats in this way? This could have major impact on observed RNA expression.
Various reports state that individual housing induces stress response in rats and changes in various parameters including (but not limited to):
Variations in biochemical (Perez, Canal, Dominguez et al 1997) and immunological (Baldwin, Wilcox and Baylosis 1995) measurements,
Behavioural changes consistent with social deprivation (such as reduced mobility, increased tail chasing and self-grooming) (Hurst, Barnard, Nevison et al 1997)
The development of an abnormal gait in animals housed in social isolation from
weaning (Roberts, Clarke and Greene 2001).
Therefore, the recomendations are: "Rats should not be housed individually unless ... rats should be able to be in visual, auditory and olfactory contact with other rats." (https://www.animalethics.org.au/)

B) Statistical analysis (line 155). Variance of measurements within groups (standard deviation) is required to better frame the results throughout the manuscript.

C) Study power calcultion is missing. It would give another perspective on presented findings.

Validity of the findings

A) lines 131, 184, figure 3. What kind of evidence were used in construction protein-protein interactions? That should be disclosed in methods. I would suggest disabling "interactions" based on "text mining" evidence.

---

## Round 0.2 · accepted · Accept

I believe that the authors have followed all suggestions of the Reviewers. Thus the paper can be accepted.